# Soluble Urokinase Plasminogen Activator Receptor: A Promising Biomarker for Mortality Prediction Among Critical ED Patients

**DOI:** 10.3390/ijms26041609

**Published:** 2025-02-13

**Authors:** Piotr Wozniak, Mariusz Sieminski, Jan Pyrzowski, Rafael Petrosjan, Jakub Głogowski-Kulasza, Jakub Leszczyński-Czeczatka

**Affiliations:** Department of Emergency Medicine, Medical University of Gdansk, 80-210 Gdansk, Poland; pwozniak@gumed.edu.pl (P.W.); jan.pyrzowski@gumed.edu.pl (J.P.); r.petrosjan@gumed.edu.pl (R.P.); jgkulasza@gumed.edu.pl (J.G.-K.); jczeczatka@gumed.edu.pl (J.L.-C.)

**Keywords:** suPAR, emergency department, mortality risk, in-hospital mortality, inflammatory markers

## Abstract

Patients admitted to the emergency department (ED) are a highly diverse group in terms of the risk of death. In overcrowded EDs, it becomes crucial to quickly and reliably estimate the risk of death or significant health deterioration. For this purpose, the concentration of soluble urokinase plasminogen activator receptor (suPAR) in plasma has been studied in recent years in various patient populations. In the present study, we tested the hypothesis that measuring suPAR upon the ED admission of critically ill patients can identify those at the highest mortality risk. To verify this hypothesis, we analyzed the relationship between suPAR plasma concentration, other biochemical parameters, and Early Warning Scores (EWSs) on admission and survival to hospital discharge. The study group consisted of 61 ED patients with priority 1 in the Manchester Triage System (MTS), excluding patients with illness caused by environmental factors. Positive correlations between suPAR and inflammatory parameters such as CRP and PCT, as well as the warning scales MEWS, MEDS, and qSOFA, were confirmed. Plasma suPAR concentration on admission was found to be a promising predictor of in-hospital mortality. The study indicated the potential prognostic value of suPAR as the mortality risk predictor for a specific population of critically ill ED patients.

## 1. Introduction

Overcrowding in emergency departments (EDs) is a growing phenomenon in many countries worldwide. This issue, among many negative effects, leads to an increased incidence of medical errors, higher patient mortality, and a greater risk of preventable deaths [1,2,3]. Several factors contribute to ED overcrowding, including inefficiencies in other parts of healthcare systems, an aging population, and heightened public expectations for quick diagnostics and treatment [3]. ED overcrowding also overloads medical personnel working under increased pressure, resulting in secondary financial losses for hospitals and national economies. Addressing the problem of ED overcrowding is a complex and challenging task that involves both the internal organization of hospital and ED operations and external factors. Simply increasing the number of beds in the ED does not solve the problem [4,5]. For these reasons, the earliest possible selection of ED patients based on their risk and severity of prognosis, as well as the subsequent adjustment of the required intervention’s timing and nature, is of paramount importance. Soluble urokinase plasminogen activator receptor (suPAR) is a soluble form of the membrane-bound receptor uPAR (CD87), primarily expressed in immune system cells such as neutrophils, activated T-lymphocytes, macrophages, endothelial cells, and smooth muscle cells. The uPAR protein, composed of three domains (D-I, D-II, D-III), is anchored to the cell membrane via glycosylphosphatidylinositol (GPI). The binding of its ligand, uPA facilitates the detachment of the receptor, resulting in the formation of soluble uPAR (suPAR) [6,7]. This process is promoted by inflammatory reactions, making suPAR levels reflective of the body’s immune activation [6]. In the following years, multiple studies demonstrated that suPAR is associated with various chronic diseases, including cardiovascular, liver, kidney, and lung diseases [8,9,10,11,12]. Studies have shown that elevated suPAR levels predict the development of diseases such as cancer in the general population [6,12,13,14,15,16]. Elevated suPAR levels are also associated with the course of various infectious diseases, including tuberculosis, HIV, malaria, sepsis, meningitis, and pneumonia, including COVID-19 [8,17,18]. suPAR levels are high in critically ill patients regardless of the primary cause of their condition [17,19,20,21,22]. This is not surprising, as this group of patients often exhibits non-specific immune system activation [23]. The normal suPAR level in plasma is below 4 ng/mL in healthy individuals, around 4–6 ng/mL in unselected ED patients, and above 6 ng/mL in critically ill patients [9,22]. Measuring suPAR levels is useful for supporting clinical decisions and triage in the ED [24,25]. A low suPAR level indicates a good prognosis and supports patient discharge, while a high suPAR level indicates disease presence, progression, and severity, warranting further clinical attention. Incorporating suPAR measurement into ED triage has been shown to enhance ED throughput and reduce costs associated with unnecessary hospitalizations by identifying up to 22% more patients who can be safely discharged [26]. Therefore, suPAR serves as a marker of mortality risk and health deterioration, as well as a predictor of survival to hospital discharge, and it can also be used to monitor disease progression. The suPAR levels in blood are stable, not subject to diurnal variation, and unaffected by meals. The baseline individual suPAR level increases with age [27]. suPAR can be measured in various bodily fluids, most commonly blood serum and EDTA plasma [27]. It has been demonstrated that suPAR levels remain stable even after the prolonged storage of serum and plasma samples under deep freezing conditions; they can also survive multiple freeze–thaw cycles [28]. suPAR is also relatively easy to measure using ELISA, immunoturbidimetric tests, and lateral flow devices for point-of-care testing (POCT), which was utilized in this study. While suPAR plasma concentrations among ED patients have been studied widely, there is still not enough data concerning the subgroup of the most urgent ED patients.

This study aims to verify the utility of suPAR, comparing it to regular, easily accessible biochemical and physiological parameters for early in-hospital mortality risk prediction in critically ill ED patients. The analysis includes routinely measured biochemical parameters, calculable risk scales, and the suPAR plasma concentration. The study tests hypotheses regarding the relationships between biochemical parameters, risk scales, and hospital discharge survival, as well as the interactions among these parameters, particularly focusing on suPAR levels. Risk scales with proven prognostic value for hospital mortality include the NEWS, MEWS, MEDS, Shock Index (SI), the modified SI (mSI), SIRS, and qSOFA. Among the routinely measured biochemical parameters in ED plasma, inflammatory markers (CRP, PCT) were studied.

## 2. Results

### 2.1. Characteristics of the Study Group

#### 2.1.1. Demographic Structure of the Tested Population

Initially, the study group consisted of 75 patients. Before the statistical analysis, 14 patients were excluded due to qualification errors or a lack of essential data that would prevent analysis. The remaining 61 patients, aged between 34 and 91 years (creating a mean age of 66.98 years), of whom 68.9% were men and 31.1% were women, were subjected to statistical analysis.

#### 2.1.2. Physiological Parameters Measured During Triage and Calculated EWSs

Among the triage parameters, an impaired level of consciousness was observed in 82% of patients (GCS < 15 points), with a median GCS of 9 points (range 3–15, IQR = 5) for the study group, indicating deep disturbances of consciousness. Other triage parameters of the study group are summarized in Table 1.

#### 2.1.3. Risk Scores Calculated

SIRS ≥ 2 points was found in 57.4% of the study population, making up 35 patients. The average SIRS value for the study group was 1.85 points. A MEDS scale score ≥ 16 points was observed in 18 people, which is 29.5% of the study group, with a median MEDS score of 13 points. All calculated risk scores parameters are summarized in Table 2.

#### 2.1.4. suPAR

Plasma suPAR levels ≥ 6 ng/mL were found in 46 patients, corresponding to 75.4% of patients, with a mean suPAR level of 10.34 ng/mL and a standard deviation of 4.63 ng/mL. The median suPAR value in the study group was 12.1 ng/mL (Table 3; Figure 1).

#### 2.1.5. Hospitalization Data—In-Hospital Mortality

52.5% of patients died during the analyzed hospitalization, including 12 patients (19.7% of the study population) who died during their ED stay. Conversely, 47.5% of the study group survived hospital discharge (Table 4).

### 2.2. Results Analysis

#### 2.2.1. Death Prior to Hospital Discharge

CRP

In the analysis of the relationship between CRP concentration and the risk of in-hospital death, a Chi-square test was used to compare CRP ≤ 50 mg/L against CRP > 50 mg/L. The analysis showed a relationship between the variables X2(1) = 4.77; *p* = 0.028 (Figure 2).

PCT

The relationship between plasma procalcitonin (PCT) concentration at ED admission and the risk of in-hospital death was analyzed for ranges of PCT ≤ 0.5 ng/mL and PCT > 0.5 ng/mL. A Chi-square test was used. The analysis showed a relationship between the variables X2(1) = 5.33; *p* = 0.021. In the group of patients with PCT > 0.5 ng/mL, the in-hospital death rate was 45.4% higher compared to the group with PCT ≤ 0.5 ng/mL. Using the non-parametric Mann–Whitney U test, the analysis did not show a relationship between the variables Z = 0.36; *p* > 0.05 (Figure 3).

NEWS

The studied population was divided into two groups depending on the NEWS score. A Chi-square test was conducted. The analysis did not show a relationship between the variables X2(1) = 0.26; *p* = 0.608 and, additionally, did not find a heightened in-hospital death risk in the group with NEWS > 6 points or NEWS ≤ 6 points.

MEWS

A Chi-square test was used. The analysis did not show a relationship between the variables X2(1) = 1.49; *p* = 0.221. Belonging to the group with a MEWS score > 4 points compared to MEWS ≤ 4 points was not associated with a significantly higher in-hospital death risk.

MEDS

The relationship between in-hospital death frequency and belonging to the group with a MEDS score > 16 points and the group with MEDS ≤ 16 points was examined. A Chi-square test was applied. The analysis did not show a relationship between the variables X2(1) = 2.31; *p* = 0.129.

SI

The study population was divided into a group with SI > 0.7 and subgroups with SI > 1.3 and SI ≤ 0.7. A Chi-square test was used. The analysis did not show a relationship between the variables X2(1) = 0.68; *p* = 0.411. Belonging to any of the above groups was not associated with a higher death risk.

mSI

A Chi-square test was used. The analysis did not show a relationship between the variables X2(1) = 0.11; *p* = 0.740. No significant relationship with in-hospital mortality was found in the subgroup with mSI > 1 compared to the subgroup with mSI ≤ 1.

SIRS

The study population was divided into a subgroup with SIRS < 2 and SIRS ≥ 2. A Chi-square test was applied. The analysis did not show a relationship between the variables X2(1) = 0.03; *p* = 0.852. Belonging to the studied SIRS ranges was not related to the recorded in-hospital mortality.

qSOFA

The study population was divided into a subgroup with qSOFA < 2 and qSOFA ≥ 2. A Chi-square test was used. The analysis did not show a relationship between the variables X2(1) = 0.18; *p* = 0.666. The qSOFA score < 2 compared to qSOFA ≥ 2 did not reflect a significant difference in the in-hospital mortality risk in the study group.

suPAR

Initial plasma suPAR concentration during admission was analyzed in terms of its association with in-hospital mortality, comparing suPAR < 6 ng/mL, suPAR ≥ 6 ng/mL, and, additionally, for ranges below and above the median, suPAR < 12.1 ng/mL and suPAR ≥ 12.1 ng/mL. The analysis showed a borderline trend association between suPAR ≥ 6 ng/mL and increased risk of in-hospital death compared to the group with suPAR < 6 ng/mL, X2(1) = 2.92; *p* = 0.056, with a more pronounced connection for the group above the median suPAR ≥ 12.1 ng/mL, X2(1) = 5.90; *p* = 0.015 compared to the group with suPAR < 12.1 ng/mL. The parametric Student’s *t*-test for independent samples showed a relationship between the variables t(59) = 2.91; *p* < 0.05. A significantly higher suPAR concentration was found in patients who died during the analyzed hospitalization (Figure 4).

#### 2.2.2. Multivariate Model Analysis

We attempted to fit our data into a multivariate linear logistic regression model using variables with *p* < 0.1 in univariate testing (listed in Table 5) along with age and gender as predictors. We used the fitglm function of MATLAB. However, neither the model (*p* = 0.59) nor any of the predictors were found to be significant (see respective *p*-values listed in Table 5). Our analysis was limited by a relatively large proportion of missing simultaneous measurements of CRP and PCT where, in many cases, only one of these parameters was assessed (*n* = 20/61 subjects). Removing one or the other of the above predictors from the analysis did not improve the results. We interpret these findings as an indication of a need to extend the study to a larger patient population (Table 6).

#### 2.2.3. suPAR Negative and Positive Predictive Value

The NPV and PPV calculated for in-hospital mortality prediction are summarized in Table 7.

#### 2.2.4. Receiver Operating Characteristic (ROC)

To assess how the cutoff point influences the sensitivity and specificity of various predictors of in-hospital death, the Receiver Operating Characteristic (ROC) curve and the area under the curve (AUROC) values were calculated. The method showed good diagnostic performance (AUROC > 0.7) for suPAR and PCT only (Table 8; Figure 5). Of note, suPAR slightly outperformed PCT in the high sensitivity regime, achieving, in its optimal “elbow” point, around 80% sensitivity at 60% specificity. The optimal performance of PCT was in turn around 70% sensitivity at 85% specificity.

## 3. Discussion

Predicting a patient’s mortality risk in the days following their admission to the ED and hospital, essentially predicting the failure of hospital treatment, is a widely researched parameter in the global literature [29,30,31,32,33,34]. In our study, we found a positive correlation between in-hospital mortality and the levels of CRP, PCT, and suPAR in plasma. No statistically significant relationship was found between any of the analyzed warning scales (NEWS, MEWS, MEDS, SI, mSI, SIRS, or qSOFA) and in-hospital mortality, which, though intriguing, in view of the existing literature (particularly regarding the NEWS and MEWS scales), should be treated carefully and considered to be potentially related to the size and characteristics of our study group [32,35,36].

Considering the relationships found for abnormally high CRP and PCT plasma concentration and increased risk of in-hospital mortality, it should be noted that the majority of patients in the study group had aforementioned parameters that were above the normal range, which should be associated with the prevalence of sepsis among the diagnoses made. Only 12 individuals, or almost 21% of patients (N = 58), were within the normal range for CRP, while 19 patients, representing 46% of the group (N = 41), had PCT levels within the normal range. In the available literature, PCT and CRP as prognostic factors for in-hospital mortality have been most frequently analyzed for populations composed solely of patients diagnosed with sepsis, adding a preference for their serial assessment. The results of our study are consistent with the results of these trials, particularly in regard to the application of procalcitonin [31,37,38,39].

The prognostic value of plasma suPAR concentration upon admission to the emergency department, concerning in-hospital mortality, was analyzed in this study through four methods. Using a Chi-squared test, the relationship between suPAR concentrations ≥ 6 ng/mL and an increased risk of death during hospitalization compared to the group with suPAR concentrations < 6 ng/mL was found to be on the edge of statistical tendency due to the very uneven distribution of the study group. A significantly clearer relationship was found for the group above the median suPAR ≥ 12.1 ng/mL compared to the group with suPAR concentrations < 12.1 ng/mL. This is the result of high mean suPAR values in the studied population, arising from the specificity of the patient selection, which comprised patients with the highest triage priority. A similar suPAR value of 12 ng/mL has already been successfully studied for the ED population with suspected sepsis [40]. Statistically significantly higher suPAR concentrations were found in patients who died during the studied hospitalization. The prognostic value of suPAR and PCT concerning mortality during hospitalization was also independently confirmed using the ROC curve, obtaining AUROC values of 0.7047 and 0.7185, respectively. Despite the lack of statistical significance demonstrated in the multivariate analysis, the obtained association of suPAR with in-hospital mortality for critical-state ED patients is promising and warrants further investigation. Of note, our results were coherent with other ED populations [9,17,20,24,26,33,41].

### 3.1. Study Limitations

The main limitation of this study is the relatively small research group. This limitation directly prevented the performance of some analyses and statistical interpretations, making others less relevant. As the study was conducted during the challenging period of the COVID-19 pandemic, the recruitment period for the study group and data collection were interrupted by surges in COVID-19 cases, during which the work of the ED, as the site of the study, was subject to extraordinary rules and priorities that temporarily hindered the collection of the research data. Nevertheless, interesting results were obtained in a similar study conducted during the same period on a similarly small group of patients [42].

Although only three patients in the study group were diagnosed with COVID-19, the potential bias of this infection on suPAR levels should be considered. A recent study found that patients with COVID-19 had higher mean plasma suPAR levels than non-COVID-19 septic patients [42]. Among other limitations noted during the interpretation of the results, it is important to mention the lack of complete data collected in the preliminary stage regarding CRP, PCT, and the parameters of multiple organ failure, which precluded the incorporation of the SOFA scale into the study.

#### Statistical Method Limitations

Our results indicate that a multivariate analysis would best be performed on a larger and more homogeneous population and that the statistical power of the current study was sufficient only for investigating preliminary associations.

## 4. Materials and Methods

### 4.1. Study Location

The study was conducted at the Emergency Department, University Clinical Center, Medical University of Gdansk, Poland. It is a 1100-bed university hospital with 35,000 ED visits per year. It is one of 4 emergency departments localized within Tricity metropoly (Gdańsk, Gdynia, Sopot in Poland), inhabited by 1,000,000 citizens.

### 4.2. Study Group of Patients

The study included adult patients in the ED, both women and men, categorized as the highest risk group upon admission according to the MTS triage scale (priority 1, immediate, red). Patients with a known traumatic or environmental cause of illness (codes S00–Y98, according to ICD10), patients who consciously refused to consent to participate in the study, patients under 18 years of age, and incapacitated patients were excluded from the study. The study group consisted of all patients meeting the inclusion criteria during a non-continuous study period from 1 November 2019, to 30 June 2021, interrupted by circumstances preventing the study, e.g., waves of the COVID-19 epidemic. The study group was not pre-limited and was closed after enrolling 75 patients.

### 4.3. Manchester Triage System and Warning Scales

To assign a priority within the Triage Scale (MTS), it is necessary to know the patient’s main complaint or reason for presenting to the Emergency Department as well as selected physiological parameters. Among the measured parameters are the level of consciousness assessed using the Glasgow Coma Scale (GCS), respiratory rate (RR), peripheral oxygen saturation (SpO_2_), blood pressure (BP), and heart rate (HR). The MTS is based on the initial assumption of the life-threatening condition of each patient and, consequently, focuses on the early steps that pose an immediate threat to life, the confirmation of which qualifies the patient for priority 1, indicating a need for immediate medical assistance. If the assessment proceeds without confirmation of specific discriminators and relevant symptoms associated with the type of condition, the urgency of the examination is reduced in a stepwise manner until it reaches priority 5 (awaiting, blue). Priority 1 MTS, which is a preliminary condition for qualification for this study group, regardless of the type of illness, includes a universal set of symptoms such as severe consciousness disturbances with an acute onset, obstruction or threat to the upper airway patency, shock or cardiac arrest. In addition to the above universal picture of a priority 1 MTS patient, depending on the dominant reason for admission, the MTS also takes into account more specific symptoms, such as seizures, significant hemorrhage, etc. In the ED, at the admission stage, supplementary to the MTS and based on the vital parameters measured within the MTS framework, additional risk scales, such as the NEWS and SIRS, are automatically calculated. The values of these scales do not affect the triage priority defined within the MTS but provide an important warning function within the MTS priorities. Since the study group consisted exclusively of patients with the highest MTS priority, physiological parameters in patients requiring immediate life-saving procedures were measured simultaneously with emergency actions.

### 4.4. Sample Collection Method for Laboratory Parameter Testing

As part of the preliminary diagnostic and therapeutic procedures, standard samples of peripheral venous or arterial blood are taken from all patients in life-threatening conditions. A regular laboratory parameters panel includes the inflammatory parameters CRP and PCT. Additionally, for patients meeting the inclusion criteria for the study, a blood sample was taken into a 10 mL EDTA tube to determine the suPAR concentration.

### 4.5. Method of Sample Collection and suPAR Measurement

For patients meeting the inclusion criteria for the study, in addition to the standard blood samples for laboratory tests, an additional 10 mL of blood was collected into an EDTA tube without the need for another venipuncture. This sample was then centrifuged for 15 min at a force of 5000× *g* in a separate room in the Emergency Department to separate the plasma. The obtained plasma was then divided into two halves using an automatic pipette, with one part being used for immediate testing and the other being frozen at −70 °C for possible future testing. Trained Emergency Department staff carried out quantitative measurements of suPAR levels in the plasma using a POCT device (suPARnostic QUICK TRIAGE, Virogates, Birkerød, Denmark), employing the lateral flow method with a testing range 0 to 15 ng/mL. After about 20 min, the result was recorded in a database, which was an anonymized list of patients participating in the study, stored on a designated computer in a spreadsheet format.

### 4.6. Retrospective Data Completion on Hospitalization Course

After the completion of hospitalization for each patient participating in the study, a trained research assistant updated the computerized database with missing information from the hospital information system. For this purpose, they searched for the necessary laboratory test results for analysis, performed automated calculations of other clinical scales (MEWS, MEDS, SI, mSI, qSOFA), and documented the treatment course and outcome, highlighting patients who did not survive until hospital discharge. In cases of technical failure of suPAR levels measurement at the ED admission, which occurred in 14 instances, the measurement was repeated using the frozen portion of the plasma collected upon admission.

### 4.7. Patient Consent for the Study Participation and Approval from the Bioethics Committee

Patients meeting the qualifying criteria were asked to provide informed consent to participate in the study by filling out an informed consent form, which was part of the study. The attempt to obtain patient consent was made upon ED admission, before any actions related to the study were initiated, in the case of conscious and oriented patients. In the case of unconscious patients, or those unable to give informed consent at the time of admission, who constituted the majority of the study group, another attempt to obtain consent was made with a delay in the days following hospitalization if the patient’s state of consciousness improved sufficiently. In cases of refusal to consent to participate in the study, or where previously given consent was withdrawn, the identifiers and measured parameters of the patient would be removed from the list of study participants, though such instances were not recorded. Approval for the study was obtained from the Independent Bioethics Committee for Scientific Research at the Gdańsk Medical University on 4 March 2019.

### 4.8. Statistical Methodology

All statistical calculations were performed using MATLAB lic. No R2024a statistical package. Qualitative variables were presented using counts and percentage values. Quantitative variables were characterized using mean/standard deviation or median/interquartile range. The significance of differences between groups was assessed using Chi-square, Mann–Whitney U, and Student’s *t*-tests. For a multivariate analysis, a logistic regression analysis was applied using variables associated with *p* < 0.1 in univariate tests. Age and gender were also included as covariates. The crosstabulation performed for Chi-square testing applied commonly used or previously published cutoff values. For suPAR, the negative predictive value (NPV) and positive predictive value (PPV) were also calculated. The effect of threshold variation was additionally addressed using a ROC/AUROC analysis.

## 5. Conclusions

The concentration of the suPAR protein in plasma has the potential for prognostic significance in assessing the probability of the in-hospital mortality of patients admitted in critical condition to the emergency department. The concentration of suPAR in plasma upon hospital admission turned out to be one of strongest predictors of in-hospital mortality in the studied group. This fact was demonstrated using several different statistical tests as well as the ROC curve. Although it requires verification in further studies in larger patient populations, the plasma suPAR concentration appeared to be more useful than the EWS scales (NEWS, MEWS, MEDS, SI, and mSI) studied in this context. In conclusion, a single measurement of plasma suPAR concentration at the time of ED admission has a potentially high prognostic value in the identification of patients most at risk of in-hospital mortality. Further studies on larger patient populations, including multicenter studies, are needed to verify and establish the results of the current study.

## Figures and Tables

**Figure 1 ijms-26-01609-f001:**
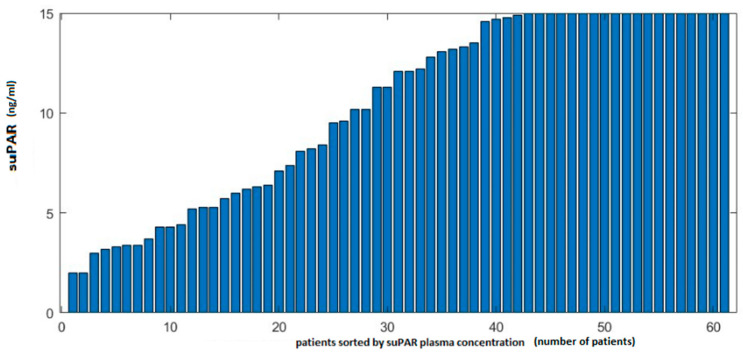
Distribution of suPAR concentration values in the study group. Testing range 0 to 15 ng/mL.

**Figure 2 ijms-26-01609-f002:**
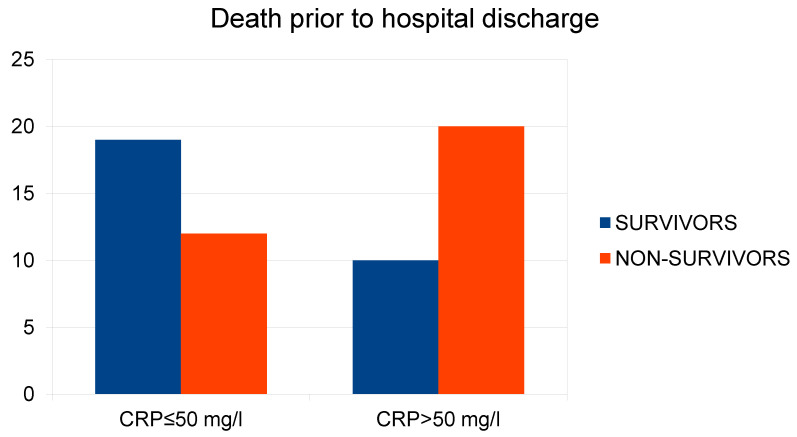
Relationship between plasma CRP concentration upon ED admission and the in-hospital mortality rate.

**Figure 3 ijms-26-01609-f003:**
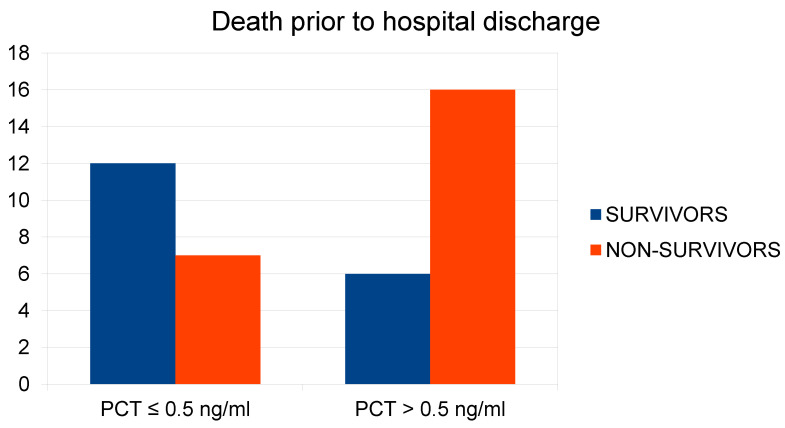
Relationship between plasma PCT concentration at hospital admission and mortality rate during the studied hospitalization.

**Figure 4 ijms-26-01609-f004:**
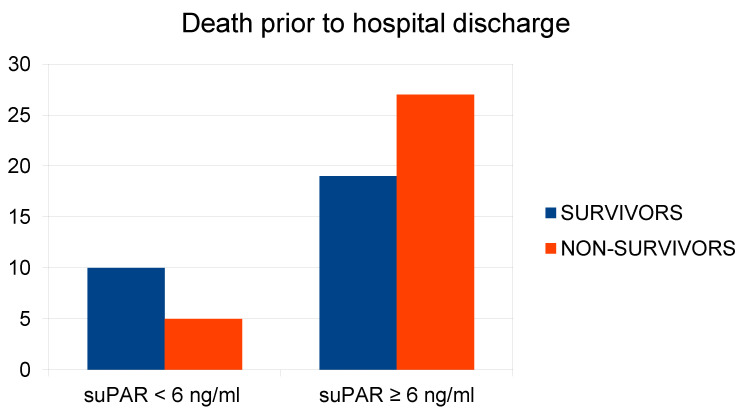
Relationship between plasma suPAR concentration at hospital admission and in-hospital mortality rate.

**Figure 5 ijms-26-01609-f005:**
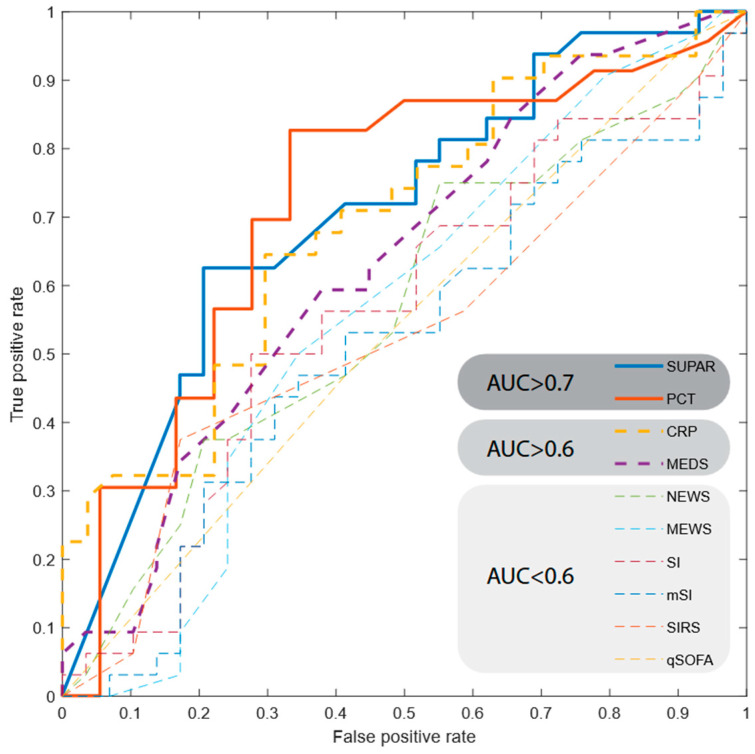
Receiver Operating Characteristic (ROC) curves for suPAR and other predictors of in-hospital mortality.

**Table 1 ijms-26-01609-t001:** Summary of the values of physiological parameters measured during triage in the study group.

*TRIAGE Physiological Parameters*	*N*	*Min*	*Max*	*M*	*SD*
SpO_2_—Oxygen saturation (%)	60	65	100	91.28	7.95
SBP—Systolic blood pressure (mmHg)	61	47	220	114.03	37.01
DBP—Diastolic blood pressure (mmHg)	61	20	102	66.16	19.33
MAP—Mean arterial pressure (mmHg)	61	29.7	138.6	81.30	24.15
HR—Heart rate (min.^−1^)	61	40	156	95.00	25.12
RR—Respiratory rate (min.^−1^)	61	10	33	19.38	6.28
Body temperature (°C)	61	29	41	36.79	1.56

**Table 2 ijms-26-01609-t002:** Summary of the scores of the study group in early warning scales and risk scales. N—group size; IQR—interquartile range.

*Quick Sepsis-Related Organ Failure Assessment*	*N*	*Min*	*Max*	*Median*	*IQR*
** *qSOFA* **	61	0	3	1	1.00
** *National Early Warning Score, Modified Early Warning Score* **					
**NEWS**	61	1	17	10	6.00
**MEWS**	61	0	12	4	3.00
** *Systemic Inflammatory Response Syndrome* **					
** *SIRS* **	61	0	4	2	2.00
** *Mortality in Emergency Department Sepsis* **					
** *MEDS* **	61	3.00	23.00	13	8.25

**Table 3 ijms-26-01609-t003:** Values of suPAR concentration in plasma in the study group. N—group size; M—mean; SD—standard deviation.

	*N*	*Min*	*Max*	*M*	*SD*
**suPAR** (ng/mL)	61	2	15	10.34	4.63

**Table 4 ijms-26-01609-t004:** In-hospital mortality in the study group. N—group size.

	*N*	*%*
**In-hospital mortality**	**32**	**52.5**
**Survival to hospital discharge**	**29**	**47.5**

**Table 5 ijms-26-01609-t005:** Summary of the statistical analysis of potential mortality predictors. *p*—significance level.

*suPAR* (ng/mL)	*Death Prior to Hospital Discharge*		*p*
	*No*	*Yes*	
suPAR < 6	10	5	* 0.056 *
suPAR ≥ 6	19	27	
suPAR < 12.1	19	11	* 0.015 *
SuPAR ≥ 12.1	10	21	
***CRP* (mg/L)**			
CRP ≤ 50	19	12	* 0.028 *
CRP > 50	10	20	
***PCT* (ng/mL)**			
PCT ≤ 0.5	12	7	* 0.021 *
PCT > 0.5	6	16	
** *NEWS* **			
NEWS ≤ 6	7	6	*0.608*
NEWS > 6	22	26	
** *MEWS* **			
MEWS ≤ 4	19	16	*0.221*
MEWS > 4	10	16	
** *MEDS* **			
MEDS ≤ 16	24	21	*0.129*
MEDS > 16	5	11	
** *SI* **			
SI ≤ 0.7	12	10	*0.411*
SI > 0.7	17	22	
** *mSI* **			
mSI ≤ 1	13	13	*0.740*
mSI > 1	16	19	
** *SIRS* **			
SIRS < 2	12	14	*0.852*
SIRS ≥ 2	17	18	
** *qSOFA* **			
qSOFA < 2	17	17	*0.666*
qSOFA ≥ 2	12	15	

**Table 6 ijms-26-01609-t006:** Univariate and multivariate model analysis results.

	Age	Sex	suPAR	CRP	PCT	NEWS	MEWS	MEDS	SI	mSI	SIRS	qSOFA
*p*-value in univariate model	0.667	0.794	0.005	0.012	0.018	0.376	0.412	0.070	0.431	0.873	0.668	0.597
*p*-value in multivariate model			0.184	0.147	0.553			0.443				

**Table 7 ijms-26-01609-t007:** Number of patients who survived to hospital discharge or died during hospitalization in relation to serum suPAR concentration. NPV—negative predictive ratio; PPV—positive predictive ratio.

	suPAR ≥ 6 ng/mL	suPAR < 6 ng/mL
survive to hospital discharge	19	10
In-hospital death	27	5
**PPV6 = 0.5870**		
**NPV6 = 0.6667**		
	**suPAR ≥ 12.1 ng/mL**	**suPAR < 12.1 ng/mL**
survive to hospital discharge	10	19
In-hospital death	21	11
**PPV12 = 0.6774**		
**NPV12 = 0.6333**		

**Table 8 ijms-26-01609-t008:** Summary of the utility of selected scales, indicators, and suPAR concentration in predicting in-hospital mortality. AUROC—area under the ROC curve.

Predictor	AUROC
NEWS	0.5663
MEWS	0.5609
MEDS	0.6347
SI	0.5593
mSI	0.5124
suPAR	0.7047
CRP	0.6923
PCT	0.7185

## Data Availability

Anonymized database can be seen at: https://docs.google.com/spreadsheets/d/1b9Ul1rhTxJFysf9dJC6l6V6xHaK7gmW0/edit?usp=sharing&ouid=104562029595175553256&rtpof=true&sd=true (accessed on 14 December 2024).

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
