# Peer review of "Soluble Urokinase Plasminogen Activator Receptor: A Promising Biomarker for Mortality Prediction Among Critical ED Patients"

_ijms, 2025, doi:10.3390/ijms26041609_

Round 1
Reviewer 1 Report (Previous Reviewer 3)
Comments and Suggestions for Authors
suPAR a valuable biomarker for mortality prediction among critical ED patients
The authors present an analysis of plasma suPAR as a prognostic indicator for mortality. The paper is well-written and covers an interesting topic but the statistical treatment is inconsistent and needs further work in order to be robust enough to be published.
Comments
Could the fact that "testing 142 range 0 to 15 ng/ml" be added to the legend of Figure 1? Also, the y-axis needs units (ng/ml).
In Table 5, I am not sure that the p-values are the most helpful way to think about the problem. We could also think about the problem as a test for likely death, to prioritise the "most in need". In this case, the 2x2 matrices could be thought of as true positives, true negatives, false positives, false negatives, and the sensitivity and specificity of the prognostic test, and confidence intervals could be calculated around these estimates. To me, the CRP test > 5 produces quite a lot of false positives, but also does the best job of identifying the true positives (28), so it has the highest sensitivity. Isn't that the most important thing?
CRP performs worse on AUROC, but only marginally, and again, isn't sensitivity the most important thing?
On the multivariarate analysis, there is no explanation in Methods of what approach was used. This needs to be explained. Why not try a simple PLS-DA model in MetaboAnalyst, where a .csv file can be uploaded under "statistical analysis, one factor" with concentrations of the different indicators, sample IDs in column 1, death status in column 2, and then the concentrations in the remaining columns (samples in rows). The data can then be processed to use an algorithm like PLS-DA to test whether a panel of 2-3 plasma features (CRP, SuPAR) might perform better than a single indicator.
https://dev.metaboanalyst.ca/MetaboAnalyst/upload/StatUploadView.xhtml
Alternatively, the authors could construct a logistic regression model for death versus no-death, and derive the feature importance for each variable (suPAR, CRP, etc etc). A logistic regression model would also allow the calculation of adjusted odds ratios, by including age / sex (one-hot encoded) as covariates.
Finally, the 3D bar charts do not make the data easier to read, actually make the visualization harder to read against the axes. Suggest to make these 2D and remove the numbers above each bar (in 2D it will be easier to read the axes).
Author Response
Comment 1:
Could the fact that "testing 142 range 0 to 15 ng/ml" be added to the legend of Figure 1? Also, the y-axis needs units (ng/ml).
Answer 1: Thank you for advice, the change was made as suggested.
Comment 2:
In Table 5, I am not sure that the p-values are the most helpful way to think about the problem. We could also think about the problem as a test for likely death, to prioritise the "most in need". In this case, the 2x2 matrices could be thought of as true positives, true negatives, false positives, false negatives, and the sensitivity and specificity of the prognostic test, and confidence intervals could be calculated around these estimates. To me, the CRP test > 5 produces quite a lot of false positives, but also does the best job of identifying the true positives (28), so it has the highest sensitivity. Isn't that the most important thing?
Answer 2: Suggested statistical changes were made. The discussion concerning CRP value was added.
Comment 3:
On the multivariarate analysis, there is no explanation in Methods of what approach was used. This needs to be explained.
Answer 3: explanation has been added to the Methods section.
Comment 4:
Why not try a simple PLS-DA model in MetaboAnalyst, where a .csv file can be uploaded under "statistical analysis, one factor" with concentrations of the different indicators, sample IDs in column 1, death status in column 2, and then the concentrations in the remaining columns (samples in rows). The data can then be processed to use an algorithm like PLS-DA to test whether a panel of 2-3 plasma features (CRP, SuPAR) might perform better than a single indicator.
https://dev.metaboanalyst.ca/MetaboAnalyst/upload/StatUploadView.xhtml
Alternatively, the authors could construct a logistic regression model for death versus no-death, and derive the feature importance for each variable (suPAR, CRP, etc etc). A logistic regression model would also allow the calculation of adjusted odds ratios, by including age / sex (one-hot encoded) as covariates.
Answer 4: The suggested logistic regression model was applied.
Comment 5:
Finally, the 3D bar charts do not make the data easier to read, actually make the visualization harder to read against the axes. Suggest to make these 2D and remove the numbers above each bar (in 2D it will be easier to read the axes).
Answer 5: Thank you for this valuable input. The mentioned graphs have been changed to 2D.
Reviewer 2 Report (Previous Reviewer 2)
Comments and Suggestions for Authors
The minor corrections are acceptable
Author Response
Thank you very much for your contribution and acceptance.
Reviewer 3 Report (New Reviewer)
Comments and Suggestions for Authors
In my opinion, this paper does not have a high impact, as it has a small sample (group of patients), without being relative to a specific disease. No comorbidities are mentioned which are strongly related with the prediction efficiency of the biomarkers. In addition, the analyses conducted did not provide any important (significant) results.
Author Response
Comment:
In my opinion, this paper does not have a high impact, as it has a small sample (group of patients), without being relative to a specific disease. No comorbidities are mentioned which are strongly related with the prediction efficiency of the biomarkers. In addition, the analyses conducted did not provide any important (significant) results.
Answer: Understanding the above point of view, we believe that the paper is valuable. It is currently the only study in the world examining the relationship between suPAR plasma concentration at ED admission and patient mortality for a unique group of patients with the highest priority for ED admission (MTS :1). The study is pilot in nature and of course requires expanding the study group.
The exclusion of comorbid diseases in our study is due to the use of suPAR and other studied predictors as parameters supporting TRIAGE and determining patient prognosis at a very early stage of admission to the ED, when we often do not have complete data about the patient.
Moreover, the available literature regarding suPAR for other patient groups indicates that comorbidities that increase plasma suPAR concentration also increase the risk of patient death to a similar extent.
Reviewer 4 Report (New Reviewer)
Comments and Suggestions for Authors
Dear authors,
I read with interest your paper as SUpar is a potential "triage tool" in the ED. However, I have 3 concerns about your paper
1. There is no explanation of the selection of 75 patients to be included. With the drop-out, you end up with 61 patients.
2. I read basic statistical errors. In quantitative data, you have continous and non-continuous data. Of non-continuous data, such as GCS, you may not calculate means. A GCS of 9.18 does not exist. Medians can be used and Q25-Q75.
3. There is an overpresentation of data. For instance, table 3 & 4 have no added value. Also the introduction and discussion could be shorter.
Author Response
Comment 1:
There is no explanation of the selection of 75 patients to be included. With the drop-out, you end up with 61 patients.
Answer 1: Thank you for this remark. As noted in section 2.2 “Study Group of Patients”: The study group was not initially limited and was closed after enrolling 75 patients. This limitation was introduced due to the pilot nature of the study. Expansion of the study group is planned in the future continuation of the study.
Comment 2:
I read basic statistical errors. In quantitative data, you have continous and non-continuous data. Of non-continuous data, such as GCS, you may not calculate means. A GCS of 9.18 does not exist. Medians can be used and Q25-Q75.
Answer: Thank you for this very valuable comment. The suggested changes regarding the median and quartiles have been done.
Comment 3:
There is an overpresentation of data. For instance, table 3 & 4 have no added value. Also the introduction and discussion could be shorter.
Answer 3: The data presentation and discussion have been modified. However, we believe that additional informations and different methods of statistical approach are valuable, because the relatively small study group makes the analysis more challenging.
Round 2
Reviewer 1 Report (Previous Reviewer 3)
Comments and Suggestions for Authors
The authors have responded to my comments thoroughly, which is very much appreciated. The results presented in the work are preliminary in nature, but all limitations are appropriately flagged and discussed for the reader, and in my view the work will make a contribution to the literature.
Author Response
Thank you very much for all your comments and recommendations during the review process. We truly value your input into our study. Best regards.
Reviewer 3 Report (New Reviewer)
Comments and Suggestions for Authors
Dear Authors,
thank you so much for the revised paper. The changes on the paper give a better understanding on your study but again, in my opinion no significant results are presented apart from SuPAR - 12.1 relationship with deaths prior discharge. The problem is that the SuPAR 12.1 is an index that it valid only for your study as it includes the median of your group (12.1). The same index could not be valid and used as decision support information in the future because the next group's characteristics (median) might be different and may do not lead to the same relation. In addition, when a patient is admitted, you do not know the "median of the sample". So, my concern is if the 12.1 will always be a 12.1 to use it as a prediction variable...
Another point is the chi-square result of the CRP. One cell has less than 5 observations (3 patients), so I am not sure that the p-value (0.027) can be accepted as valid and reliable result.
Finally, I do not know why your performed a regression analysis with all the variables, even with those which are not statistical significant (or close to 0.05) during the 1-1 comparison? You could not have better results on regression when 1-1 comparisons are not confirmed.
If your paper will be finally accepted please try to accommodate the above.
Best Regards
Author Response
Comment 1:
in my opinion no significant results are presented apart from SuPAR - 12.1 relationship with deaths prior discharge. The problem is that the SuPAR 12.1 is an index that it valid only for your study as it includes the median of your group (12.1). The same index could not be valid and used as decision support information in the future because the next group's characteristics (median) might be different and may do not lead to the same relation. In addition, when a patient is admitted, you do not know the "median of the sample". So, my concern is if the 12.1 will always be a 12.1 to use it as a prediction variable...
Response 1:
Thank you for this comment.
The above study is unique because it examines a specific subgroup of patients in the most critical condition admitted to the ED. Studies of this group in the ED setting are generally difficult and rare. Our study is valuable because, to our knowledge, it is the first publication examining suPAR in the highest priority TRIAGE group (MTS-1). So far, the prognostic significance of plasma suPAR levels for ED patients has been studied mainly in an unselected group of ED patients.
Nevertheless, when comparing the results of existing studies examining suPAR plasma concentrations in other groups of ED patients at increased mortality risk, our threshold value of suPAR of 12 ng/ml was similar. This publication was cited in our study:
https://pubmed.ncbi.nlm.nih.gov/38321472/
In summary, a plasma suPAR level of 12 ng/ml can be postulated as decision support for a specific subgroup of patients admitted to the ED in severe condition. These results obviously require confirmation in further studies on larger groups of patients.
Comment 2:
Another point is the chi-square result of the CRP. One cell has less than 5 observations (3 patients), so I am not sure that the p-value (0.027) can be accepted as valid and reliable result.
Response 2:
Thank you for this valuable input. We changed the cutoff of CRP to 50 mg/l which populated all the bins more adequately and was still statistically significant.
Comment 3:
Finally, I do not know why your performed a regression analysis with all the variables, even with those which are not statistical significant (or close to 0.05) during the 1-1 comparison? You could not have better results on regression when 1-1 comparisons are not confirmed.
Response 3:
Multivariate analysis was modified according to your suggestions.
We sincerely thank you for all the valuable feedback and comments you provided on our study during the peer review process. We believe that it is now much better thanks to them.
Round 3
Reviewer 3 Report (New Reviewer)
Comments and Suggestions for Authors
The current paper can be published although that the study has limited sample.
This manuscript is a resubmission of an earlier submission. The following is a list of the peer review reports and author responses from that submission.
Round 1
Reviewer 1 Report
Comments and Suggestions for Authors
Unfortunately, the manuscript has not been significantly improved and the serious issues I highlighted in my first review report are yet to be addressed.
Reviewer 2 Report
Comments and Suggestions for Authors
You need to fix formating of your tables where there should be decimals there are commas. The comments I made on first draft have been addressed
cheers
Reviewer 3 Report
Comments and Suggestions for Authors
suPAR a valuable biomarker for mortality prediction among critical ED patients
The authors present an analysis of plasma suPAR as a prognostic indicator for mortality. The paper is well-written and covers an interesting topic but the statistical treatment is inconsistent and needs further work in order to be robust enough to be published.
Major points
Figure 1 looks a little odd. Was the maximum recordable value 15 (i.e. the data are capped at 15 but the "true" value may have been higher, or did a large number of participants have a coincidental suPAR concentration of exactly 15 by coincidence? This information could usefully be added to the legend. Also, the y-axis needs units.
The comparison between suPAR, CRP, PCT etc in Table 6 and the text is partial. For all indicators, a cut-off threshold is used. For suPAR only, the median value is additionally presented. Please can the authors show the median value (calculated the same way) for all the other indicators in order that a fair comparison be made.
As a more general point, I am not sure that the p-values are the most helpful way to think about the problem. We could also think about the problem as a test for likely death, to prioritise the "most in need". In this case, the 2x2 matrices could be thought of as true positives, true negatives, false positives, false negatives, and the sensitivity and specificity of the prognostic test, and confidence intervals could be calculated around these estimates. To me, the CRP test > 5 produces quite a lot of false positives, but also does the best job of identifying the true positives. Isn't that the most important thing?
Why are CRP and PCT excluded from the AUROC analysis (Table 7, Figures 5-10)? Being cynical, it makes me wonder if their AUROC was better than suPAR plasma concentration.
Line 365, "Using the Chi-squared test, the relationship between suPAR concentrations ≥ 6 ng/ml and an increased risk of death during hospitalization compared to the group with suPAR concentrations < 6 ng/ml was found to be on the edge of statistical tendency, due to the very uneven distribution of the study group." This is your belief, but it is not proper statistical analysis. The result was not statistically significant. Would it have been statistically significant if you had a more even distribution? Maybe it would, but we cannot tell, and you present no evidence for this.
Line 379, please add the comparison for CRP.
Line 402, there is nothing to stop the authors from constructing a logistic regression model for death versus no-death, and deriving the feature importance for each variable (suPAR, CRP, etc etc). A logistic regression model would also allow the calculation of adjusted odds ratios, by including age / sex (one-hot encoded) as covariates.
Minor points
I think SIRS is first presented at line 84, so this is where the acronym should be spelled out
Figures 2 and 3, a 3D bar chart does not make the data easier to read, actually makes the visualization harder to read against the axes. Suggest to make these 2D and remove the numbers above each bar (in 2D it will be easier to read the axes).
Comments on the Quality of English Language
English is fine and did not hinder my understanding of the manuscript.